# ERK Activity Imaging During Migration of Living Cells In Vitro and In Vivo

**DOI:** 10.3390/ijms20030679

**Published:** 2019-02-05

**Authors:** Eishu Hirata, Etsuko Kiyokawa

**Affiliations:** 1Division of Tumor Cell Biology and Bioimaging, Cancer Research Institute of Kanazawa University, Ishikawa 920-1192, Japan; ehirata@staff.kanazawa-u.ac.jp; 2Department of Oncologic Pathology, School of Medicine, Kanazawa Medical University, Ishikawa 920-0293, Japan

**Keywords:** biosensor, FRET, cell migration, ERK

## Abstract

Extracellular signal-regulated kinase (ERK) is a major downstream factor of the EGFR-RAS-RAF signalling pathway, and thus the role of ERK in cell growth has been widely examined. The development of biosensors based on fluorescent proteins has enabled us to measure ERK activities in living cells, both after growth factor stimulation and in its absence. Long-term imaging unexpectedly revealed the oscillative activation of ERK in an epithelial sheet or a cyst in vitro. Studies using transgenic mice expressing the ERK biosensor have revealed inhomogeneous ERK activities among various cell species. In vivo Förster (or fluorescence) resonance energy transfer (FRET) imaging shed light on a novel role of ERK in cell migration. Neutrophils and epithelial cells in various organs such as intestine, skin, lung and bladder showed spatio-temporally different cell dynamics and ERK activities. Experiments using inhibitors confirmed that ERK activities are required for various pathological responses, including epithelial repair after injuries, inflammation, and niche formation of cancer metastasis. In conclusion, biosensors for ERK will be powerful and valuable tools to investigate the roles of ERK in situ.

## 1. Introduction

Cells receive extracellular stimuli and transduce signals for growth, differentiation, and migration. Various types of receptors including receptor tyrosine kinases (RTKs), G protein-coupled receptors, cytokine receptors, and cell-substrate receptors change their conformation upon ligand binding and recruit downstream proteins. A representative example is the growth factor signalling pathway. This is initiated upon the engagement of growth factors (GFs) with a RTK, which recruits adaptor proteins and guanine nucleotide exchange factors (GEFs) to activate the RAS small GTPases. In turn, activated RAS triggers a phosphorylation cascade consisting of RAF-MEK-ERK (extracellular signal-regulated kinase) [1]. In this cascade, active phosphorylated forms of the upstream kinases directly phosphorylate the downstream kinases: RAF phosphorylates MEK, which then phosphorylates ERK. Finally, activated ERK phosphorylates various proteins, including cell-cycle regulating transcription factors such as ETS and AP1 [2], apoptosis regulating BCL-2 family proteins [3], and proteins related to cell motility [4,5,6,7]. Since activating mutations in RTKs, RAS, RAF, and MEK occur in diverse types of human cancer, inhibiting ERK, their downstream effector, would appear an effective strategy for cancer intervention. To determine the levels of activated ERK in different biological and pathological environments, researchers have stained various cells and tissues using antibodies against phosphorylated ERK. The limitation of this method is that it does not provide spatio-temporal information of ERK activities in living cells. The discovery of green fluorescent protein (GFP) has led to major progress in live imaging, including the development of many biosensors for ERK activities in living cells and tissues. In this review, we first describe the latest information on fluorescent protein-based biosensors for ERK activity, before we summarize the findings on cell migration derived using these biosensors in vitro and in vivo.

## 2. ERK Biosensors

### 2.1. FRET-Based Biosensors

Förster (or fluorescence) resonance energy transfer (FRET) is a nonradiative energy transfer process between two fluorophores located in close proximity, and its efficiency is strongly dependent on the distance and orientation between the fluorophores [8]. The first GFP-based FRET biosensor, Cameleon, was established to observe calcium ion concentrations in living cells [9]. In this biosensor, calmodulin and calmodulin-binding peptide M13 are placed between two FPs, a blue or cyan fluorescent protein (BFP or CFP) and a green or yellow fluorescent protein (GFP or YFP) (Figure 1A). Binding of Ca^2+^ to calmodulin induces a conformational change of the biosensor and increases energy transfer from B/CFP (a donor FP) to G/YFP (an acceptor FP), resulting in enhanced emission of 535 nm light from the biosensor. Based on this principle, the Raichu, a FRET biosensor that reports Ras activity, was developed [10]. Later, FRET biosensors for various protein kinase activity were developed [11].

#### 2.1.1. EKAR-Based FRET Biosensors

The first FRET biosensor for ERK, EKAR, was developed by Harvey et al. [12] (Figure 1B). The original version of EKAR contained GFP (a donor FP), WW domain of human cdc25, an ERK substrate peptide fused with a docking domain and red FP (an acceptor FP) in this order. When activated, ERK phosphorylates the threonine in the substrate peptide (PDVPRTPVGK), which is recognized by the WW domain in the biosensor. Later, Komatsu et al. changed the FP pair to CFP/YFP [13]. They also modified the biosensors by using a longer linker (116 aa, EV-linker), which improved the sensitivity and dynamic range of the biosensor (Figure 1B). 

Various fluorescent proteins have been discovered or modified to improve the imaging quality in vitro and in vivo, and these proteins have been applied to the EKAR-EV biosensor. A novel method for the reversible exchange of the heterodimeric partners of green and red dimerization-dependent fluorescent proteins (ddFPs) has been established [14]. ddFPs are a series of fluorogenic protein pairs comprising a fluorogenic FP-A and a non-fluorescent FP-B, wherein dimerization results in increased FP-A fluorescence. The CFP/YFP pair of EKAR-EV was replaced with ddFPs, the RPF-A/B pair generating RAB-EKARev [15]. 

Another modification of EKAR-EV was brought about by the super resonance energy-accepting chromoprotein (sREACh) [16]. REACh, a dark YFP-based resonance energy-accepting chromoprotein, is a YFP variant with extremely small quantum efficiency but high absorbance, and its variant super REACh (sREACh) was developed with point mutations [17]. Both REACh and sREACh are useful as FLIM (fluorescence lifetime imaging microscopy) acceptors [17]. There are two methods to measure the FRET efficiency. One is the ratiometric method, in which the ratio between the donor and acceptor fluorescence intensities is calculated. Since it requires only minor changes to the optical path and filters of the common microscope, ratiometric measurement is widely used in biology studies. The other method is measurement of the fluorescence lifetime of the donor [18]. Since the lifetime is shortened as the FRET efficiency increases, the lifetime of the donor can be used as a readout of FRET. Unlike in ratiometric measurement, the brightness of the acceptor is not important in the lifetime measurement, because FRET–FLIM measures only the donor fluorescence. Instead, a high acceptor absorption coefficient is required for high FRET efficiency. To realize this goal, the YPet/ECFP pair of EKAR-EV was replaced with sREACh/EGFP to generate EKARsg. Further mutations were added in sREACh to generate sREAChet, and the resulting EKARet biosensor carrying the sREAChet/EGFP pair is currently the best FLIM-biosensor available [16] (Figure 1C). This biosensor has been introduced to an organotypic hippocampal slice culture of mouse pups, allowing visualization of the ERK activity in individual spines of a neuron. Readers will find various improved fluorescent biosensors including EKARs at the Fluorescent Biosensor Database (https://biosensordb.ucsd.edu/index.php).

#### 2.1.2. Miu2 FRET Biosensor

While the EKAR-based biosensors detect the phosphorylation ability of ERK, another FRET-based biosensor, Miu2 (MAPK indicator unit ERK2), reflects the conformational change of ERK2, and the localization of ERK in the cells [19] (Figure 1D). The Miu2 probe comprises YFP, ERK2, and CFP from the N-terminus. When MEK is overexpressed, the Miu2 biosensor is located in the nucleus. In the absence of MEK overexpression, Miu2 is localized mainly in the cytoplasm. Three to 10 minutes after epidermal growth factor (EGF) stimulation, Miu2 is translocated to the nucleus in HeLa cells. The ERK activations detected by Miu2 in the nucleus and the cytoplasm show similar kinetics. The same study revealed that the time-course of ERK activation detected by Miu2 and Western blotting using a phosphor-specific antibody were similar. 

### 2.2. Kinase Translocation Reporters

Ratiometric FRET biosensors are a useful tool for the detection of kinase activity with high sensitivity and specificity. Their drawback is that two colours are used for one kinase activity; this limits the monitoring of other signalling pathways simultaneously. To overcome this point, kinase translocation reporters (KTRs), which are multicolour biosensors for multiple kinase activity, have been developed [20]. In these biosensors, a kinase docking site, a regulatory nuclear localization signal (NLS) peptide, a kinase phosphorylation substrate and a regulatory nuclear export signal (NES) peptide are fused in this order and labelled with a FP at its C-terminus. When a kinase of interest is activated, it binds to the biosensor, phosphorylates the substrate and masks the regulatory NLS, resulting in an increased shuttling of the biosensor from the nucleus to the cytoplasm. To exploit this phenomenon, the ERK-KTR reporter, in which Elk (aa 312-356) and FQFPS were used as the docking site, was constructed (Figure 1E). Using a similar strategy, the JNK and p38 kinase activities were quantified in a single living cell [20]. The nucleus, cell cycle indicator Geminin, ERK-KTR, and AKT-KTR from a single vector were all successfully imaged within a single cell. Since the ERK and Phosphatidyl Inositol 3 kinase (PI-3K)-AKT pathways are both downstream of RAS and are known to have different spatio-temporal dynamics, visualization of the ERK and AKT pathways is a powerful tool to investigate the signalling events starting from various growth factors in the individual cells [21]. 

### 2.3. FIRE

An ERK activity reporter called FIRE (Fra-1-based integrative reporter of ERK), which consists of mVenus [22], a bright YFP variant, fused to the PEST domain of Fra-1, was recently developed [23] (Figure 1F). This report is based on the observation that the PEST domain of Fra-1 is phosphorylated by ERK and stabilized. By measuring the fluorescent intensity of FIRE, the activity of ERK is measured.

## 3. ERK Activities In Vitro

### 3.1. Pulse Activation of ERK for Proliferation

By expressing the ERK biosensors in cultured cells on a plastic/glass dish, the pulsatile ERK activity (at least 10 to 20 pulses in a single cell cycle) has been visualized in MCF10A, a human mammary epithelial cell line [23]. In this experiment, low concentrations of EGF induced isolated bursts lasting 20–30 min (mean 27 min) were interspersed by dormant periods of 1–4 h, while high concentration pulses increased in duration and decreased in spacing; in other words the ERK activity became sustained. Since the variations in ERK pulse activity strongly influence the proliferative activity of individual cells, the authors proposed the novel concept of “frequency modulation”. According to this concept, it is the frequency rather than the amplitude of the discrete pulses that responds to and encodes the strength of the extracellular stimulus [24]. Similar ERK activity pulses (8–20 frequency/day), but without growth factor stimulation, have also been observed in normal rat kidney epithelial NRK-52E cells [25]. In this report, the density of the cells was the key to determining the pulse frequency. At a certain density of cells, spontaneous firing of ERK activity occurred and ERK activity was propagated to adjacent cells through the cleavage of EGF-family ligands on the cell surface by a disintegrin and metalloproteases (ADAMs). Importantly, in an experiment realizing light-switchable ERK activation by recruiting RAF to the plasma membrane, pulsatile ERK activation, but not sustained ERK activation, induced SRF-regulated genes such as Fos and Egr1, leading to cell proliferation. 

### 3.2. ERK in Cell Migration In Vitro

#### 3.2.1. Pulse Activation of ERK for Migration

Both cell migration and proliferation are governed by the propagating wave of ERK activation [26]. A wound-healing assay of Madin-Darby canine kidney (MDCK) epithelial cells, a cell line derived from renal tubules, revealed two distinct types of ERK activation wave, a ‘‘tidal wave’’ from the wound, and a self-organized “spontaneous wave’’ in regions distant from the wound. In both cases, MDCK cells collectively migrated against the direction of the ERK activation wave. An ERK activation wave spatiotemporally controlled the actomyosin contraction and cell density. The technique described above, the light-switchable ERK activation, was applied to manipulate ERK activity in a pulsatile propagating manner to induce collective cell migration.

#### 3.2.2. ERK in a Cell Cluster/Cyst Rotation in 3-D Culture

One of the differences between in vitro and in vivo environments is the stiffness of the substrate, which greatly differs among tissues and organs. The hardest of these is the bone, which has the elastic module of 2–4 GPa [27], which is similar to those of glass/plastic dishes. It is also reported that the appropriate stiffness is required for each tissue to develop, differentiate, and maintain their function and morphology, and that rigid micro-environments promote cancer progression [28]. It is therefore necessary to reconstitute the in vivo tissue microenvironment when studying the morphogenesis of multicellular tissue architectures. The representative model for epithelial structure is a spherical cyst and tubular structures comprised of MDCK cells [29]. In this system, a single MDCK cell seeded on/in extracellular matrix (ECM)-rich gel grows to form a cell cluster in the early phase, and later becomes a cyst that comprises a monolayer of polarized cells surrounding a fluid-filled lumen, which is similar to the epithelial structure in the human body [30]. During the cystogenesis, RAS activity is homogenous among the cell in the cyst, but stochastic ERK activation is observed [25]. 

We recently found that K-RAS expression accelerates the rotation of cell clusters or cysts (Figure 2A) [31]. The major downstream pathways from the active K-RAS are RAF-MEK-ERK, RALGEF, and PI3K-AKT. Using mutants of the active K-RAS which transmit signals specific to individual downstream factors, the activation of RAF and RALGEF, but not PI3K, induced rotation albeit to a much lesser extent. Similarly, expression of the active form of MEK1 induced a smaller but still significant level of cell cluster rotation. The activation pattern of the ERK in the cell clusters has not been investigated, but in the cysts, ERK activation was observed homogenously in the cells in the cysts upon the active K-RAS expression [32] (Figure 2, right panels). It is therefore likely that the endogenous activation of ERK is regulated in a different way in the K-RAS active cancer cells. We found that the active K-RAS expression induced coherent rotation of cysts, with lower velocity compared to clusters [31]. Treatment with vorinostat, an HDAC inhibitor, or active β-catenin expression enhanced the rotation of cysts, but not clusters, indicating that the molecular mechanisms for rotation differ between these two structures (Figure 2A).

Since invasion is the first step of metastasis, which is the major cause of death of cancer patients, identifying the responsible proteins that promote cancer cell invasion is a topic of intense research in basic and clinical studies. In addition to single-cell migration associated with epithelial-mesenchymal transition, collective cell migration is also considered to play an important role in cancer cell invasion [33,34]. Notably, pathological studies often emphasise the importance of cell clusters, as they form a subset of collective cell migration. For example, the tumour buddings and poorly differentiated clusters (PDCs) are observed at the invasive edge in colorectal cancer (CRC) (Figure 2B). Tumour budding is defined as the formation of single cancer cell clusters of <5 cancer cells [35], while PDCs are cancer clusters composed of ≥5 cancer cells and lacking a gland-like structure [36]. Tumour budding and PDCs in CRC are strongly predictive of lymph node metastases and worse prognosis [37,38,39]. Although it is expected that the cell clusters are migrating and sequentially invade into the lymphatic vessels, no one has witnessed this process due to difficulty of the imaging in vivo. We therefore aim to employ cell clusters in vitro as a model system to elucidate the cellular and molecular mechanisms of clustered cell migration in vivo. In addition to our study, others have used MCF10A, a mammary gland cell lines, to demonstrate that the activation of ERK upregulates cell motility within a mature MCF10A cyst, though without increased invasiveness [40]. Similar rotation of cell cluster/cyst primary and cultured breast cancer cells have also been observed in vitro [41,42]. It is also possible that the rotation of clusters is unrelated to the migration of the clusters in the ECM. Indeed, none of the cell clusters in vitro showed invasive motility toward surrounding ECM. Since chemokines secreted from the stroma, such as vessels, fibroblasts, or macrophages, were absent in our in vitro system, the clusters rotated in the gel instead of migrating. Further in vivo imaging will be needed to certify that this in vitro system is useful for collective cell migration. The candidate signalling events revealed by in vitro analysis should next be validated using clinic-pathological materials. However, staining the surgical specimen of the cancers using phospho-specific antibodies is not feasible. Probably due to this limitation, there has been no report showing that ERK activation occurs in the cell clusters/cysts at the invasive front of cancers, or that such activation is related to the prognosis.

## 4. ERK Activities in Cell Migration In Vivo

To visualize the ERK activities in vivo, we need to have mice expressing the biosensors as well as a two-photon excitation microscope [8]. In the two-photon excitation process, a fluorescent molecule simultaneously absorbs two photons with approximately twice the wavelength and half the excitation energy of a single photon [43]. In this way, high-resolution imaging becomes possible in the deep parts of living tissue using a two-photon microscope. Among the potential biosensors, transgenic (TG) mice expressing the EKAR-EV biosensor have been generated to elucidate the role of ERK activities in health and diseases [44,45]. EKAR-EV is further tagged with NES or NLS to visualize the ERK activities in the cytoplasm or nucleus, respectively. During the course of our experiments, we found that there was no difference in dynamics between ERK activities in the nucleus and cytoplasm, although EKAR-EV-NLS was suitable to track the cell migration, while EKAR-EV-NES was better for visualizing the cell shape. One caveat is that expression of the fluorescent proteins, including EKAR-EV FRET biosensors, is gradually silenced in the intestinal epithelium during aging, due to the DNA methylation of the actin promotor [46]. The flox lines of mice, in which FRET biosensors are expressed by a Cre-recombinase dependent manner, were created [47]. By crossing these flox mice with various tissue-specific or cell-specific Cre expressing mice, ERK activities in the cells of interest were visualized [47,48]. 

### 4.1. Neutrophil Migration upon LPS Treatment

The immune system consists of a variety of cell types. By using cell-specific fluorescent labelling with antibodies conjugated with chemical dyes or transgenic techniques, individual cell dynamics have been visualized in the mouse body with multiphoton excitation microscopes [49]. In FRET-TG mice, neutrophils could be easily distinguished by the segmented nuclei from lymphocytes or macrophages [44]. We first found that neutrophils migrate fastest among the cells in inflamed subcutaneous tissue, and that ERK activity in the neutrophils is correlated with migration velocity [44]. This observation was reasonable, since the neutrophils migrate to the inflamed lesion in the acute phase of inflammation, and thus they are considered a part of the first line of immune defences. Pathologists utilize the accumulation of neutrophils as a hallmark of the acute phase of inflammation in routine diagnosis. 

The neutrophils can be found in the bloodstream, with a lifespan of 6–8 h, and in tissue, where they can last up to seven days [50]. Upon inflammation, neutrophils leave the bloodstream. The recruitment processes of the neutrophils, such as rolling, adhesion, crawling, and transmigration of the venules, have been investigated in vitro and the molecular mechanisms have been identified [51,52]. We settled on a system which imaged the extravasation processes of neutrophils in the lipopolysaccharide (LPS)-treated intestine of the FRET-TG mice [53]. At the beginning of the adhesion step, when neutrophils were arrested on the endothelial cells, ERK activity remained low, and it then increased rapidly when neutrophils spread over the endothelial cells. The high ERK activity was maintained during and after crawling, transmigration, and random migration between crypts. Treatment with an MEK inhibitor reduced the ERK activities and the numbers of neutrophils crawling over the endothelial cells and migrating into the interstitial tissue. Most chemoattractant receptors expressed on neutrophils are coupled with the heterotrimeric G*_i_* protein, which activates ERK through both the α and βγ subunits of G*_i_*. Previously, it had been reported that the G*_i_*-mediated ERK activation is required for adhesion and migration of neutrophils [54], and in vivo study showed that G*_i_*-coupled BLT1, the ligand of which is LTB4, mediates ERK activation and transduces a “go” signal to neutrophils. Interestingly, when we used the FRET-TG mice to monitor protein kinase A (PKA) activity, PGE2-EP4-G-mediated PKA activation supressed ERK activity during some processes of the extravasation [53]. 

### 4.2. Myeloid-Derived Suppressor Cells (MDSCs)

To investigate the cancer cell invasion process, both cancer cells and cells in the interstitial tissue have been imaged. The 4T1 cell line, a metastatic subclone of cells derived from spontaneously arising mammary tumours from a BALB/cfC3H mouse, was broadly used as a metastatic model of breast cancer. Before injecting the 4T1 cells, bone marrow cells of a FRET-TG mouse for ERK were transferred to a recipient BALB/c mouse, and the lung, which is the major metastatic organ of this system, was observed with a two-photon excitation microscope [55]. **N**eutrophil infiltration into the lung was observed within one week after 4T1 cell inoculation. Neutrophils near the cancer cells showed activated ERK activity. Since they were positive for Ly6G/Gr1, these neutrophils were designated as granulocytic MDSCs. Knock-down of osteopontin (OPN) in 4T1 cells did not change the tumour growth, but reduced tumour metastasis. These results suggest that OPN secreted from the cancer cells recruits the neutrophils, which in turn help tumour cells to colonize the lung. OPN is a ligand for integrin and CD44, and has been reported to be elevated in the serum of cancer patients. Activation of the RAF-MEK-ERK pathway is known to be required for neutrophil extracellular trap (NET) formation, characterized by release of DNA due to chromatin decondensation and spreading [50,56]. Treatment with DNase I, which inhibits NET, suppressed colonization of 4T1 cells. It was therefore suggested that OPN-mediated ERK activation in the neutrophils mediates NET and facilitates tumour colonization.

### 4.3. Epithelial Cell Migration In Vivo

#### 4.3.1. Intestinal Epithelial Cells After Ischemic Injury

During the course of our observations, we noticed a difference in velocity among cell types; the neutrophils migrated quickly, while the epithelial cells in the intestine rarely moved over several hours of observation. Several previous studies have reported that the epithelial cells migrated as single cells or as a collective sheet in vitro [33]. In some cases, growth factors/cytokine stimulation is required for full migration velocity, and the requirement of such soluble factors is cell-context dependent. Therefore, our inability to observe the epithelial cell migration in vivo over the course of several hours may have been due to the experimental conditions being unconducive to migration—namely, inflammation may not have been a sufficiently strong stimulus for epithelial cell movement. 

Epithelial cells migrate during various biological and pathological events. Among them, we focused on the regeneration process of the intestine [48] (Figure 3). The surface of the intestine is covered by a monolayer of epithelial cells that functions as a physical barrier to protect the body from pathogens and dietary substances [57]. Ischemia induces epithelial cell injury leading to death, and after several hours to days of the injury, the integrity of epithelial cells is re-established. To model ischemic injury, we employed segmental vascular occlusion, since it induces local infarction without severe damage to the other organs. One of the mesenteric arteries near the cecum of the EKAR-EV TG mouse was occluded to block the blood supply for 50–60 min. After reperfusion, the intestine was returned to the abdomen, and the wound was closed. Twenty-four hours after ischemia, the epithelial cells were detached from the basement membrane and the crypt-villus structure was disrupted. Forty-eight hours after ischemia, monolayer epithelial cells appeared to cover the injured area. To exclusively express FRET biosensors in intestinal epithelial cells, Villin-CreERT2 TG mice [58], which express a Cre recombinase in the intestinal epithelial cells upon estrogen treatment, and Lox-STOP-lox-FRET-TG mice [47] were crossed. In those mice, higher ERK activity was observed in resealing epithelial cells in the damaged area compared to adjoining normal cells (Figure 3A). Unfortunately, we could not observe the migration of epithelial cells over several hours of observation. For skin imaging, the imaging window can be settled and observed for several days to week [59]. A similar technique can be applied to intestine imaging. However, since the mice could not live without intestinal peristalsis, and our in vivo imaging inhibited peristalsis to some extent in order to generate stable images, it was technically difficult to observe the intestine over a long period of time.

Although we could not observe the epithelial cell migration, we tried to investigate its molecular mechanisms [48]. Gefitinib, an inhibitor for EGFR kinase activity, reduced the high ERK activities in the resealing epithelial cells. Consistent with a previous report that EGFR-ERK pathway activation is mediated via upregulation of ligands for the EGF receptor (EGFR) by transcriptional activation of YAP [60], we observed that YAP was located in the resealing epithelial cells as detected by immune-fluorescence. YAP localization and the activation of ERK detected by immune-histochemistry were also observed in the surgical materials of ulcers (Figure 3B). Considering that the resealing epithelial cells in our mouse model and human surgical materials expressed E-cadherin and did not proliferate, it is plausible that they migrate as a sheet from the adjoining crypt. Our working model is that epithelial cells in the crypt adjoining the injured area start to proliferate within several hours, and cells generated during that period migrate to cover the injured area (Figure 3C). It has been established previously that rapid (within several hours) resealing of the epithelial barrier following injuries is accomplished by a process termed epithelial restitution [57,61]. Various animal models of epithelial restitution have been established, including luminal acid exposure in the rabbit duodenum [62] and colon [63], endoscopic biopsy in the mouse colon [64,65], the addition of dextran sodium sulfate to the drinking water of mice and rats [66], and irradiation of the mouse colon [67]. Since the speed and mode of epithelial repair are influenced by the cause, degree, and location of the injuries, further experiments will be needed before concluding that the signalling pathway detected in our experiments is common to all the regeneration processes in the digestive tracts. 

#### 4.3.2. Epidermis and Basal Cell Migration of the Urothelium During Wound Healing

The velocity of the basal cells in the epidermis and bladder, both of which are on the lamina propria, was measured and it revealed that the cells of the urothelium migrate faster than those of the epidermis [68]. A significant difference in mobility was not observed between the basal layer cells and the umbrella cells in the bladder. When the skin was wounded with a fine needle, epidermal cells of two to three rows from the wound edge rapidly migrated toward the wound center at a speed of 6 μm/hour. ERK activation waves were propagated from the wound edge. In the urothelium, intensive laser radiation was applied to the wound. The wound healing process was initiated soon after the laser ablation and all urothelium cells migrated at the speed 10 μm/h. Interestingly, while the MEK inhibitor inhibited the ERK activities in both cells in the epidermis and urothelium, migration was inhibited only in the epidermal cells. Treatment with dasatinib, a Src family tyrosine kinase inhibitor, reduced the migration velocity of both cell types. 

## 5. Perspective

The development of ERK biosensors and in vivo imaging techniques can be applied to various investigations of biological and pathological events in vivo. There are still, however, several limiting factors for in vivo imaging of the activities of signalling molecules. Even though the biosensors have been certified to work in vitro, it is sometimes difficult to obtain clear images in vivo. For example, the macrophages contain many auto-fluorescent proteins which interfere the specific fluorescent signals from the biosensor. Liquids, such as blood, urine, bile, and milk, may also perturb fluorescent signals with high auto-fluorescence and/or absorption of light. If the tissue contains various cells and structures with different refractive indexes, the depth of the imaging is limited. To overcome these problems, fluorescent proteins with greater brightness and longer excitation wavelength have been developed (https://www.fpbase.org/). It will also be necessary to improve the apparatus to correct coma or spherical aberrations [69,70]. Another problem is the difficulty of the long-term imaging. As with a surgical operation, several modifications of anaesthesia, temperature control, and body position, or tissue fixation apparatus with aspiration and magnet will be required [8,71].

Since ERK is located downstream of K-RAS-RAF pathway, both of which are major driving oncogenes mutated in various cancers, and the ERK1/2 inhibitor ulixertinib has been investigated as a cancer agent in phase I dose-escalation and expansion studies [72], ERK biosensors will be beneficial to cancer research. As mentioned above, the invasion and metastasis process of cancer cells require a long period to complete. There are several possible approaches for improving our understanding of cancer cells by using ERK biosensors. For example, EKAR-EV TG mice can be used to isolate cells with high activities of ERK, and then these cells can be transplanted to mice to follow the function or dynamics of ERK-activated cells [73]. By isolating the affected organoids from the EKAR-EV TG mice, the ERK activities responding to multiple inhibitor treatment can be quantified [74]. Alternatively, to model human cancers, biosensors have been introduced into both cancer cell lines and cancer-associated fibroblasts, and the effect of the inhibitors was monitored in vitro [75,76]. This could assist in drug selection or resistance of various cancers. Finally, in vitro culture of patient-derived tumor organoids (PDO) and their xenografting to nude mice (PDX) are also available [77]. Application of live cell imaging of kinase activity in PDO and PDX could contribute to reveal novel cellular dynamics and mechanisms for invasion and metastasis.

## Figures and Tables

**Figure 1 ijms-20-00679-f001:**
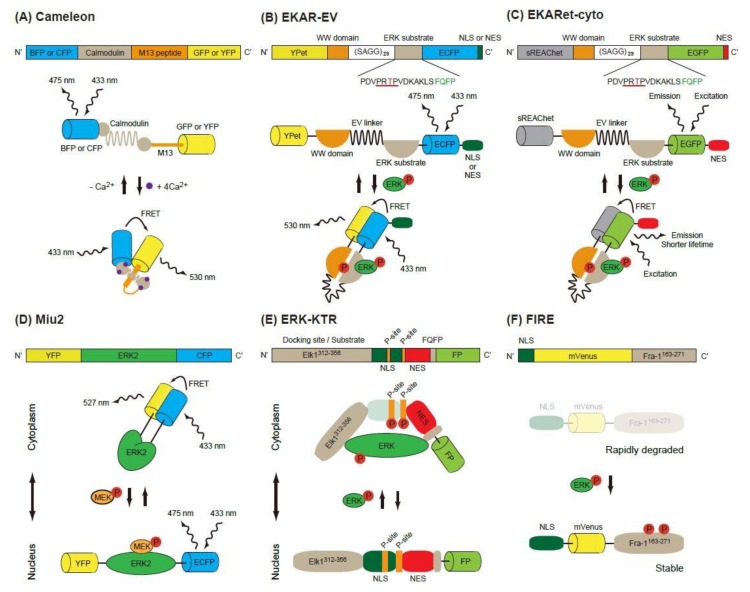
Schematic of single-molecule Förster (or fluorescence) resonance energy transfer (FRET) biosensors. (**A**) The construction and operating principle of Cameleon, a calcium biosensor. The calmodulin domain binds to M13 peptide domain in the presence of Ca^2+^, which causes a conformational change of the biosensor and increases the FRET efficiency between the donor FP (BFP or CFP) and the acceptor FP (GFP or YFP). (**B**,**C**) Similar principles are applicable in EKAR-EV (**B**) and EKARet-cyto (**C**). When phosphorylated, the threonine in the ERK substrate binds to the WW domain, which induces a conformational change of the biosensor and increases the FRET efficiency between the donor and acceptor FPs. The increased FRET efficiency can be detected as high FRET ratio (FRET/CFP) in ratiometric analysis (**B**) or shortening of the donor fluorescence lifetime in FLIM analysis (**C**). (**D**) In the Miu2 biosensor, the conformational change of ERK2 protein by active (phosphorylated) MEK results in the nuclear translocation of the biosensor and a decrease in FRET efficiency. (**E**) In the ERK-KTR biosensor, direct binding of active (phosphorylated) ERK and phosphorylation of the P-site mask the regulated NLS, which in turn causes the cytoplasmic translocation of the biosensor. (**F**) In the FIRE biosensor, phosphorylation of ERK substrate (Fra-1163-271) by active ERK stabilizes the construct and protects it from rapid degradation, thus resulting in high ERK activity and strong fluorescence signals.

**Figure 2 ijms-20-00679-f002:**
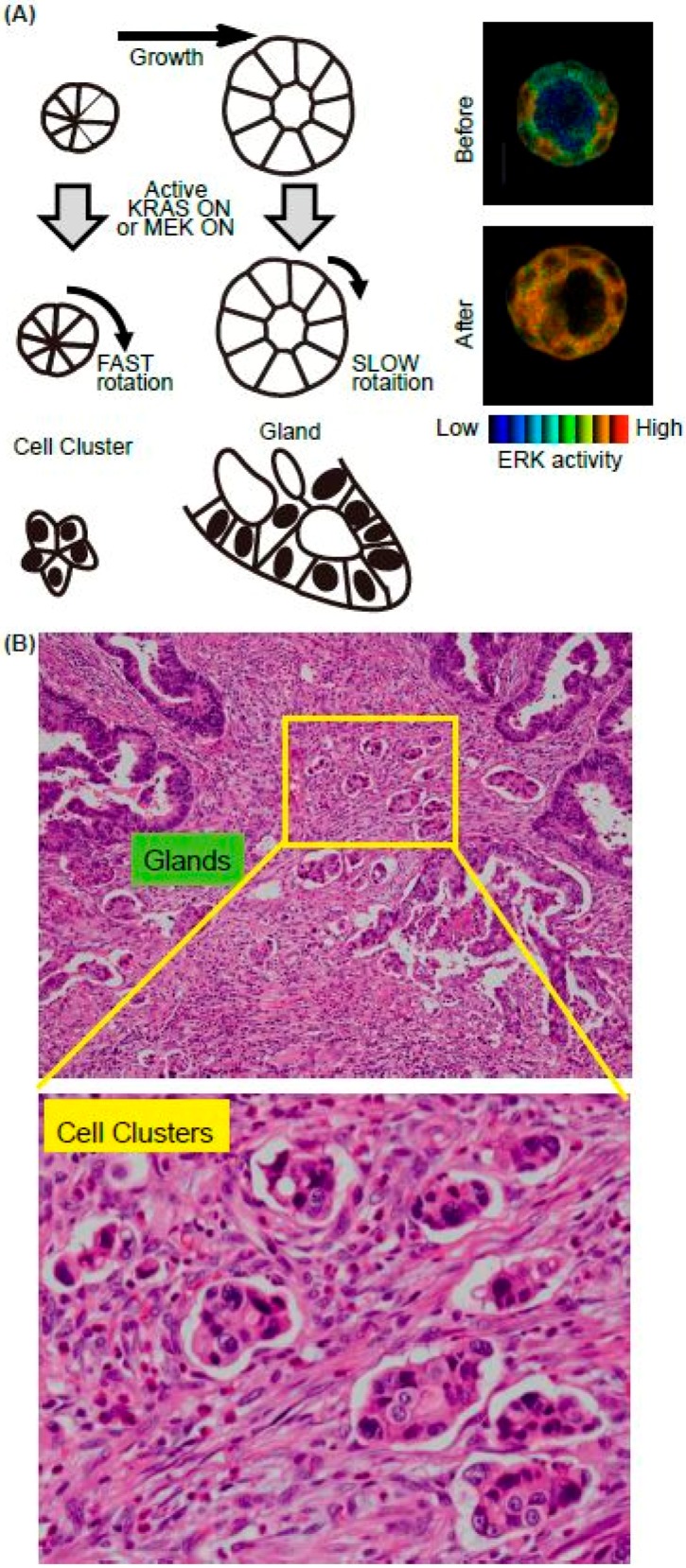
Summary of the in vitro model system for collective cell migration at the invasive front in vivo. (**A**) Schematics of the MDCK cell cluster/gland in vitro model. Upon the active K-RAS expression, ERK was activated homogenously in the mature cyst (right panels). (**B**) The representative H&E image of invasive front of CRC. Both glands and cell clusters were observed.

**Figure 3 ijms-20-00679-f003:**
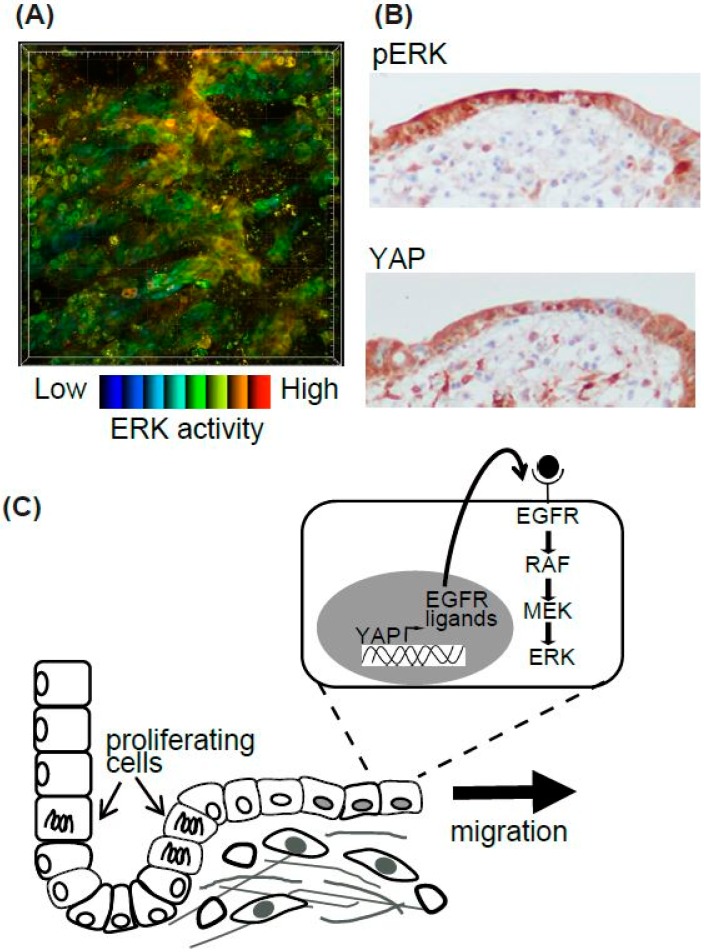
Summary of epithelial cell resealing in the intestine. (**A**) High ERK activity in the resealing epithelial cells in the EKAR-TG mouse intestine. (**B**) The representative images of pERK and YAP IHC in the human intestinal ulcer. (**C**) The working model of the resealing epithelial cells after injury.

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
