# Peer review of "ERK Activity Imaging During Migration of Living Cells In Vitro and In Vivo"

_ijms, 2019, doi:10.3390/ijms20030679_

Round 1

Reviewer 1 Report

In submitted manuscript authors described the latest information on fluorescent protein-based biosensors for ERK activity, then summarized the findings on cell migration derived using these biosensors in cells and mice. It is interesting manuscript which generally requires only minor mainly formal corrections. Here are my mainly minor comments/suggestions. Comments/suggestions: 1. I suggest to mention also ERK as a keyword, however, it is definitely up to authors. 2. Introduction section is too limited in my view. I suggest authors to mention more information about ERK, e.g. ERK signalling pathway description, downstream molecules possibly affected by this pathway, processes in which this kinase is involved, etc. I also suggest to use “in vitro and in vivo” instead of “cells and mice” at the end of introduction. 3. In Figure 1 there are missing letters “A, B, C”. Legend of this figure should be also improved, e.g. description of the respective letters (A, B, C, E…) could be also presented. 4. In chapter 2.1.2. MEK is mentioned, however, there is nothing written about this kinase in previous text. I suggest authors to include description of this kinase in introduction section among the ERK pathway description. 5. I might be better to organize abbreviations in alphabetical order. 6. References format could be more standardized, e.g. sometimes there is a comma sometimes semicolon after the year, etc. 7. Generally, formal aspect of the manuscript could be improved. Here are some of the formal mistakes I have noticed: a. Page 2, line1 – comma missing between two brackets b. Page 2, line 16 – double-space before “When” c. Page 6, line 2 – space missing before “[27]” d. Page 9, line13 – [48,52] instead of „[48][52]“ e. „in vitro“ and „in vivo“ could be written in italics within the text f. Etc.

Author Response

As attached.

Reviewer 2 Report

The authors review here current techniques and approaches on ERK activity imaging, mainly during cell migration. The topic covered is relevant and a review of the biosensors in use for ERK activity and the findings on cell migration could be of relevance for a broad audience. I enjoyed the first part of the review, were the tools available are described. However, I found that the description of the findings on ERK activity in cell migration in vitro and in vivo are too largely based on self references, and not written with a more general perspective. The language used actually reflect so and, in some sections, large parts of the text are devoted to explaining the results of a single manuscript (like on 3.2.2).

In addition:

·       Part of Figure 1 is missing. There is no A, B or C labelling.

·       Line 10. or must be deleted.

·       Vague sentences should be avoided or extended. Like for example on lines 43-44.

Author Response

As attached.

Round 2

Reviewer 2 Report

The manuscript has been improved and the changes in general respond to the concerns expressed. However, some minor text editing would be needed, mostly in the new added parts.

Also in Line 41 cures cancer is a strong statement.

Author Response

We thank the reviewe for her/his suggestion. We asked the native English speaker to edit the revised part of our manuscript. During this process, the sentence "cures cancers" is gone.